behaviour/environmental science

social dilemma, common-pool resource game, intergenerational cooperation, sustainability

**Author for correspondence:**
Chia-chen Chang
e-mails: dbschcc@nus.edu.sg, chiajen.chang@gmail.com

# Having a stake in the future and perceived population density influence intergenerational cooperation

Chia-chen Chang, Nadiah P. Kristensen,

Thi Phuong Le Nghiem, Claudia L. Y. Tan and

L. Roman Carrasco

Department of Biological Sciences, National University of Singapore, 14 Science Drive 4, Singapore 117543, Singapore

C-cC, 0000-0001-9853-8949; LRC, 0000-0002-2894-1473

Intergenerational common-pool resource games represent a new experimental paradigm in which the current generation's decision to cooperate or defect influences future generations who cannot reciprocate, providing key insights for sustainability science. We combine experimental and theoretical approaches to assess the roles of having a stake in the future (50% chance to pass the resource on to themselves in the next generation) and reminders of the presence of others (exposure to people-chatting sounds) on intergenerational cooperation. We find that, as expected, having a stake in the future increases cooperation with future generations, except when participants are also exposed to people-chatting sounds. We hypothesize that this interaction effect occurs because people-chatting sounds trigger a perception of large group size, which reduces the chance of individuals and their descendants benefiting from the pool in the future, thus reducing cooperation. Our results highlight the context-dependent effect of having a future stake on intergenerational cooperation for resource sustainability, and suggest an area of future work for environmental messaging.

## 1. Introduction

Current environmental problems (e.g. overfishing, deforestation and climate change) can be conceptualized as common-pool resource problems, where a shared resource will be degraded or destroyed if it is over-exploited [1,2]. To sustain the resource, individuals need to limit their personal use, which means forgoing short-term rewards and potentially missing out if others do not do the same. This creates a situation where each individual's short-term

self-interest is at odds with the long-term collective interest, i.e. a social dilemma [3]. Further complicating matters, environmental problems often involve a resource that is passed onto and exploited over multiple generations [4,5]. This raises the question of how to motivate and enhance cooperation in a social dilemma that is intergenerational.

One aspect that may motivate sustainable behaviour is if people have a material stake in the future of the resource. The simplest future stake is if an individual is likely to use the resource again later in life. Environmental messaging often appeals to the audience's concerns for their own future (e.g. sea-level rise [6]), and, when individuals have a long-term self-interest, they may give up short-term benefits. In the intergenerational context, environmental appeals often invoke 'future generations', which the public typically interpret as their own family and descendants [7]. If one's offspring are also likely to use the resource, then that too constitutes a material future stake. Theoretical models predict that natural selection can favour sustainable resource use if offspring are likely to use the same resource as their parents [8,9]. This theory is consistent with some observations of pre-modern societies, where the sustainable management of wild and semi-wild resources is achieved by demarcating areas to indicate family or tribal ownership [10–12]. Experiments also show that reminders of one's own children increase pro-environmental behaviour [13,14], which suggests that invoking care for one's descendants may be an effective messaging strategy. However, this strategy has been criticized on the grounds that it consolidates self-interest in one's own genealogy, whereas global environmental crises (e.g. climate change) involve a much larger scope of ethical concern, which is beyond ourselves or our offspring [7,15].

A key feature of modern environmental problems is that they take place with the presence of other people and probably under high population density, such as in cities. The presence of others can have either positive or negative effects on intergenerational cooperation. For example, even subtle cues of the presence of others can trigger reputational concerns [16], which improves cooperation [17–22] and pro-environmental behaviour (e.g. conspicuous green consumption) [17,23–25]. On the other hand, cooperation in social dilemmas typically decreases with the number of participants, even when a reputation effect is included [26], in part due to greater perceived conflict and lower perceived efficacy [27,28]. A field study also showed that urban residents depleted resources more quickly than rural residents in a dynamic common-pool resource game [29].

One important effect of the presence of others is that their behaviour can signal whether or not the resource is likely to be sustained. In a social dilemma, a participant's optimal strategy depends upon the strategies of others around them. Many environmental problems, such as increased carbon dioxide in the atmosphere, can be monitored over time, and the failure of others to behave sustainably diminishes each individual's incentive to behave sustainably themselves. If individuals perceive that the resource is unlikely to be sustained, then their best strategy is to extract the maximum possible amount of resource now before it becomes exhausted by others. This is also true in the intergenerational context, e.g. if a resource is unlikely to be passed on to one's offspring anyway, then the optimal strategy may be to maximize extraction now and pass on economic benefits instead.

A new experimental paradigm, intergenerational common-pool resource (ICPR) games, has been proposed for studying intergenerational cooperation [5,30,31] and may provide insights into how to enhance it. ICPR experiments have previously found that the resource typically deteriorates over generations unless an additional mechanism is added, such as voting or punishment [30,31]. However, these experiments have simulated generations as sequential groups of strangers and assumed the future outcomes do not influence the current generation (i.e. they do not have a future stake). Furthermore, these experiments have generally focused on spontaneous one-shot games, where there is no opportunity for a player to observe the behaviour of other players in the group. An alternative to one-shot games are dynamic games with multiple rounds, where the behaviour of others signals how quickly the resource is being depleted. It has been shown that in dynamic games, when the pool outcome will only influence people in future generations, cooperation decays rapidly over the rounds [32].

In this paper, we go beyond previous ICPR games to investigate: (i) whether a stake in the future of the pool influences cooperation; (ii) whether future stakes interact with the presence of others on influencing intergenerational cooperation; and (iii) how one-shot versus dynamic ICPR games influence intergenerational cooperation. To simulate a stake in the future of the pool, participants were told that there was a 50% chance that they would inherit the pool in the next generation of the game. This may be interpreted as a chance to use the resource again later in life, but it also represents a literal interpretation of the genealogical self-interest that is implicit in environmental appeals to care for our children's future [7,15]. To explore the effect of the presence of others, we used a sound treatment: people chatting in a coffee shop setting. The people-chatting sound has been found to trigger a sense of crowdedness and high population density [33]. Behaviour was compared with silence and nature sounds, with the latter

chosen to simulate low population density in 'rural' environments and to rule out the possible effect of sound itself. Each participant played a one-shot game and a dynamic game.

# 2. Methods

## 2.1. Participants

A total of 430 participants were recruited via an online participant recruitment portal and advertisement emails (age: 22.5 ± 2.1 years old; sex: 57.7% female; ethnicity: 87% Chinese). Participants were students from the National University of Singapore (NUS; 50% STEM majors including 27.7% having studied an environmental module). All participants gave their consent before taking part in the experiment, and they were free to withdraw from the experiment at any time. The experiment was conducted at a computer laboratory in NUS. Participants were also asked to provide demographic information, including age (open-ended), gender (female/male), ethnicity (Chinese, Malay, Indian and Others), family monthly income (collected in 10 bands, ranging from below $1000, $1000–$2999, $3000–$4999, $5000–$6999, $7000–$9999, $10 000–$12 999, $13 000–$14 999, $15 000–$16 999, $17 000–$19 999, above $20 000), whether or not they have taken environmental modules (yes/no) and their majors (open-ended).

Participants were randomly allocated to groups of five individuals who shared a common pool, resulting in 86 groups in total. The groups were then randomly split between treatments (detailed below). Half were allocated to the groups without a stake in the future of the pool (control) or groups with a stake in the future of the pool (43 groups each), and then each was further split into three sound treatments (15 groups with people-chatting sounds, 14 groups with nature sounds and 14 groups with silence).

## 2.2. Sound treatments

We exposed participants to different soundscapes during the experiment: silence, nature sounds and people-chatting sounds (a busy cafeteria). The people-chatting sounds were chosen to test the effect of the presence of others and a sense of crowdedness or high population density [33]. The nature sounds were chosen as a contrasting sound treatment to test the effects of sound itself and also to simulate low population density such as rural environments. For participants exposed to the nature sounds or people-chatting sounds, the sounds were played before any participant entered the laboratory, using speakers at the four corners of the room, at a constant 20–45 dB (list of soundtracks in electronic supplementary material, note A).

## 2.3. Intergenerational common-pool resource games

The game was designed and administered using the computer platform oTree v. 2.1.34 [34]. At the computer screens, participants spent 5–10 min reading instructions. They were given a comprehension test. Once they had answered all questions correctly, they started the games. The instructions, comprehension test and interface are included in electronic supplementary material, note B.

Each participant was informed that they were in a group of five anonymous individuals who would play a game involving multiple parts and rounds. They would extract resource points from a common pool, and each point they extracted would be converted to 50 cents (Singapore dollars). They were not told how many parts they would play, but told that one part of the game would be randomly selected and the points obtained converted to cash. They were told that their pool would be passed on to other groups and that the amount they extracted would affect subsequent groups' earnings. If their group extracted less than or equal to 50% of the pool's initial value, then the pool would be refilled to 100% of that value before passing on to the next group. However, if the percentage extracted was more than 50%, then the pool would only be refilled to 70% of its initial value. They were also told that the effect was cumulative (e.g. a repeatedly over-exploited pool would go from 100 points to 70 points to 49 points, and so on), and the pool would not generate any earnings when the pool had less than 10 points. They were told that they could be randomly allocated to any generation in the sequence; however, all participants were allocated to the first generation (100 points in total).

To compare the individual extraction behaviour with behaviour in a group where participants can learn from and adjust to others' behaviour, participants played this game as a one-shot game and then a five-round dynamic game. In the one-shot game, each individual could extract between 0 and

20 points. The maximum-profit cooperative strategy is for every participant to take 10 points (i.e. 5 players × 10 points per player = 50 points extracted). At the end of the one-shot game, participants only saw how many points they themselves extracted. After the one-shot game, participants played the five-round dynamic game, in which extractions occurred over five rounds. In each round, each individual could extract between 0 and 4 points. In the five-round dynamic game, the maximum-profit cooperative strategy is for every participant to take 2 points per round (i.e. 5 players × 5 rounds × 2 points per player per round = 50 points extracted). At the end of every round, the cumulative number of points extracted by both the participants themselves and their group as a whole was shown.

## 2.4. Future-stake treatment

We compared the resource extraction of participants between two experimental scenarios: without a stake in the future of the pool (control) versus with a stake in the future of the pool. In the control groups, participants were told that their group would not pass the same pool on to themselves in the next part of the game. In the groups with a stake in the future of the pool, participants were told that there was a 50% chance that they would pass the same pool on to themselves in the next part of the game. Taken together, the experiment follows a two (with or without a stake in the future of the pool) by three (people-chatting sounds, nature sounds, silence) experimental design.

## 2.5. Social value orientation task

After the ICPR games, participants performed a task to measure their social value orientations and were told that this task was independent from the ICPR game. The social value orientation of each individual was measured because of its potential association with cooperation and sustainable behaviour [35]. The task was the hypothetical incentivized six-item social value orientation slider measure, which asks participants to make six decisions about allocating money to themselves and another unknown person [36]. The task determined if they preferred to allocate money in one of four ways: (i) individualistically, maximizing their own benefits; (ii) altruistically, maximizing the benefits to others; (iii) prosocially, maximizing the benefits of both parties; or (iv) competitively, maximizing the difference in benefits between themselves and others.

## 2.6. Nature relatedness measurement

After the social value orientation task, participants were surveyed about their relatedness with nature. Nature relatedness was measured because it has been found to be associated with pro-environmental attitudes [37]. We used the nature relatedness scale, which measures a person's connectedness with nature as part of their identity, desire to be in nature and their concern for environmental issues from human's activities [37]. The responses to 21 statements were collected on a 5-point Likert scale (1, strongly disagree, to 5, strongly agree). The average score from each participant was then calculated, with a higher score indicating stronger nature relatedness.

## 2.7. Statistical analyses

All groups except one over-exploited the pool, so all the analyses were performed on the number of points extracted instead of the outcome. We first analysed the effect of sounds (silence versus nature versus people-chatting sounds), future-stake treatment (without versus with a stake in the future of the pool) and game version (one-shot versus five-round dynamic game) on the group extraction (i.e. total number of points extracted as a group). We used a linear mixed-effects model using the nlme package [38] in R 3.6.3 [39] with the group extraction as a response variable. Sounds, future-stake treatment, game version, and all two-way and three-way interactions were included as fixed effects. Group ID was included as a random effect. The Wald test was used to test the significance of the variables. The homogeneity of variance in the model was checked.

Similar to the group-level extraction, we also analysed the effect of sounds, future-stake treatment and game version on the individual extraction level (i.e. total number of points extracted for each individual). We used a generalized linear mixed-effects model using the lme4 package [40] with a binomial error structure. The individual extraction as a proportion of 20 points was used as a response variable. Sounds, future-stake treatment, game version, and all two-way and three-way interactions were

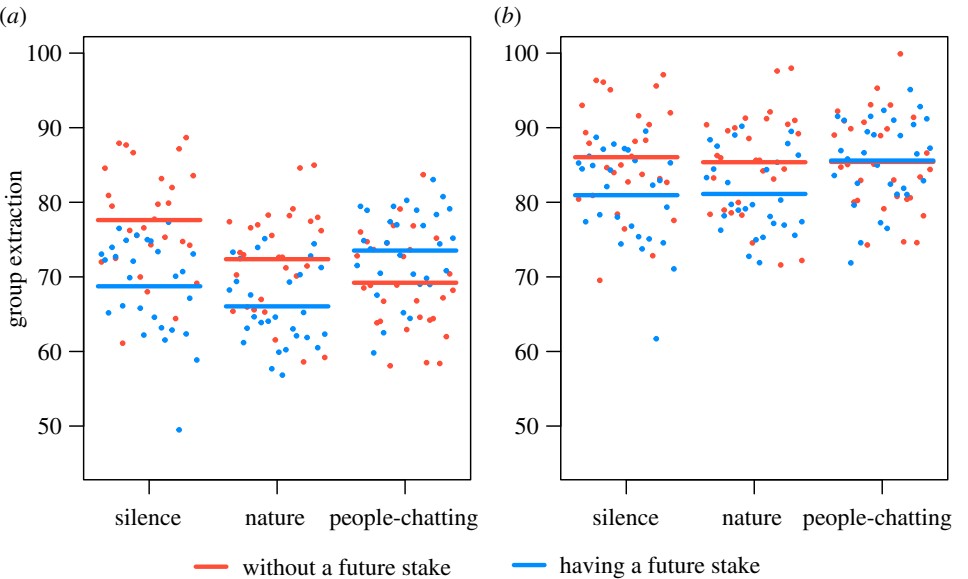

**Figure 1.** Effects of future-stake treatment (without a future stake versus having a future stake) and the sound treatment (silence, nature and people-chatting sounds) on the group extraction in a one-shot game (*a*) and five-round dynamic game (*b*). The plots include prediction lines of the best model while controlling for other variables. The visualization was done using visreg package in R.

included as fixed effects. Participant labels were nested in group ID and used as a random effect, including an observation-level random effect to account for overdispersion.

At the group and individual levels, two additional models were run. The first additional model included social value orientation (prosocial/individualistic at the individual level; number of prosocial members at the group level) as a covariate. Most individuals were categorized either as 'individualistic' or 'prosocial' except for one 'competitive' individual; therefore, this individual was excluded from the analysis. The second additional model included social value orientation and other participant's characteristics as covariates, including gender (female/male at the individual level; number of females at the group level), ethnicity (Chinese/non-Chinese at the individual level; number of non-Chinese at the group level), family monthly income (rank at the individual level; average rank at the group level), whether or not they had taken environmental modules (yes/no at the individual level; number answering yes at the group level), major (STEM/non-STEM at the individual level; number of STEM-major members at the group level) and nature relatedness (score at the individual level; average score at the group level). There was no multicollinearity between the covariates detected at both group and individual level of analyses (variance inflation factor (VIF) < 3).

## 3. Results

The model with the participant's characteristics predicted group and individual extraction better than the models without (electronic supplementary material, tables S1, and S2). In the one-shot game, approximately 46% (98 out of 215) of participants extracted 10 points or less when they had a stake in the future of the pool, while only 36% (78 out of 215) did when they had no stake in the future of the pool. Under the silent sound treatment, having a stake in the future of the pool significantly reduced resource extraction (figures 1 and 2). However, under the people-chatting sound treatment, the effect of having a stake in the future of the pool was reversed, i.e. it increased resource extraction (figures 1 and 2). In general, having a stake in the future of the pool decreased resource extraction under the silent and nature sounds treatments, but increased extraction under the people-chatting sound treatment. This interaction effect was more obvious in group extraction (figure 2*a*; electronic supplementary material, table S1, $p = 0.012$) and marginally significant in individual extraction (figure 2*b*; electronic supplementary material, table S2, $p = 0.059$).

When individuals had no stake in the future of the pool, the people-chatting sounds significantly reduced extraction when compared with silence, but this effect was only clear at the group-level analysis (figure 2). In addition, we did not detect a significant difference in extraction between

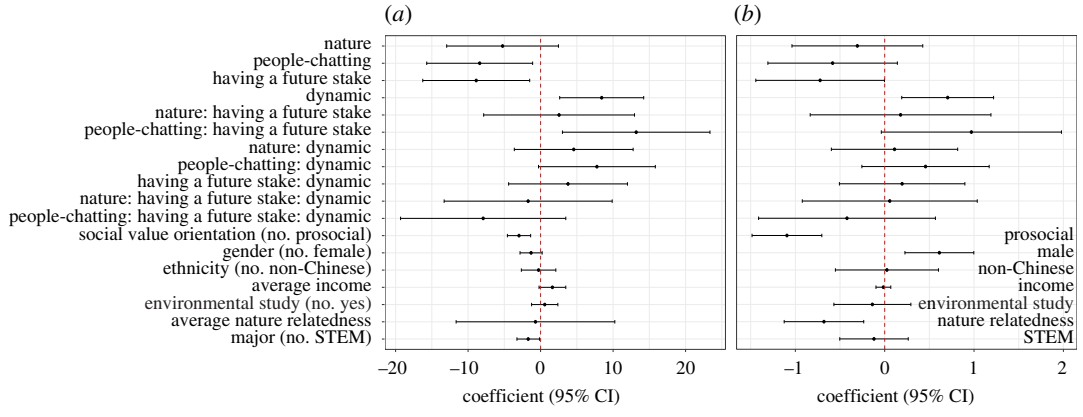

**Figure 2.** Coefficient (95% CI) of future-stake treatment (without a future stake versus having a future stake), sound treatment (silence, nature and people-chatting sounds) and game version (one-shot versus dynamic game) in group (*a*) and individual (*b*) extraction. Covariates included social value orientation, gender, ethnicity, family income, whether or not they have taken environmental modules, major and nature relatedness.

people-chatting sounds and nature sounds when there was no stake in the future of the pool (electronic supplementary material, table S3). However, when individuals had a stake in the future of the pool, people-chatting sounds reduced the positive effect of having a stake in the future of the pool, but the nature sounds did not influence its positive effect (figures 1 and 2).

Participants extracted significantly more points in the dynamic game than the one-shot game at both group and individual levels (figures 1 and 2). Only approximately 8% of the participants extracted 10 points or less (having a stake in the future of the pool: 23 out of 215; without a stake in the future of the pool: 12 out of 215).

For participant's characteristics, groups having more prosocial members extracted fewer points, and prosocial individuals also extracted fewer points (figure 2). Nature-related individuals extracted fewer points, and groups with more STEM-major members extracted fewer points (figure 2). With regard to demographic factors, we found female individuals and groups with more female members extracted fewer points (figure 2). Groups with members from higher-income family extracted slightly more points (figure 2). We did not detect a significant association between ethnicity and extraction behaviour, and we also did not detect a significant association between participants having taken environmental modules and their extraction behaviour (figure 2).

## 4. Discussion

Consistent with our expectations, we found that having a stake in the future of the pool increased cooperation. However, the positive effect that future stakes had on cooperation was diminished or reversed when participants were reminded of the presence of others (people-chatting sound treatment). We hypothesize that this negative interaction effect occurred because, although having a future stake motivated participants to forgo immediate rewards to sustain the pool for future rewards, reminders of the presence of others decreased their confidence that the pool would be sustained and, therefore, their own incentive to cooperate. This result raises the possibility that environmental messaging that (inadvertently) invokes an individual's self-interest by appealing to concern for their future self or their offspring may backfire, particularly in large-scale environmental problems involving a large number of people (e.g. climate change).

### 4.1. When there is no future stake, reminders of the presence of others could increase cooperation

When participants had no future stake in the resource pool, exposure to people-chatting sounds increased cooperation compared with silence. This positive effect was expected because people-chatting sounds invoke the presence of others [41], which may provide a subtle cue to trigger the reputation effect [16], similar to the watching eyes effect [20]. Alternatively, people-chatting sounds have also been shown to increase future orientation of individuals [33], which would also reduce the resource extraction.

We found no significant difference in extraction between people-chatting sounds and nature sounds treatments, probably because nature sounds also had a marginally positive effect on cooperation compared with silence. One possible explanation is that sound itself increases cooperation. Previous work has found that increasing the cognitive load on participants, e.g. with distracting sounds [42], causes them to rely on an intuitive cooperative behaviour [43] and promotes future-oriented and other-oriented behaviour [44,45]. An alternative explanation is that nature sounds specifically increase cooperation through a different mechanism. Combining nature sounds with a game scenario that has an obvious sustainability analogy may have activated pro-environmental values in some participants. For example, it has been found that people behave more sustainably and cooperatively after experiencing nature [46], e.g. due to reduction of materialism and increase in generosity through the experience of awe from nature [47,48], and people with higher nature relatedness also have stronger pro-environmental attitudes in general [37,49]. In support of this, we also found that more nature-related participants extracted less resources. Future studies could use soundtracks with music or white noise to isolate the effects specific to people or nature sounds from sound in general.

## 4.2. Having future stakes increases cooperation, but the reminder of presence of others undermines the positive impact

Consistent with expectations, introducing future stakes increased cooperation, but not in the people-chatting sounds treatment. There are a variety of ways that a future stake can promote sustainable behaviour. The simplest example is the one simulated in our experiment: a chance to use a resource in the future. Previous studies have also found that being future-oriented or projecting the self into the future (e.g. thinking from the perspectives of the future generations) increases intergenerational sustainability [50–52]. Another common future stake is the well-being of one's children as a form of genealogical self-interest. It is generally found that having biological children or being observed by one's own children can encourage sustainable behaviour [14,53–56]. For example, families with children reduce their energy use more in response to health and environmental information (e.g. air pollution) than families without children [53].

However, the positive effect of future stakes on cooperation was negatively affected by reminders of the presence of others. In the dynamic game, the people-chatting sounds removed the positive effect of future stakes; and in the one-shot game, people-chatting sounds reversed the effect so that people with a future stake extracted more than people without one. The negative interaction effect occurred only for the people-chatting sounds, not nature sounds, which suggests that it is specific to reminders of the presence of others and not sound exposure generally. Future studies interested in quantifying the effects of crowdedness on the interaction effect could use soundtracks that vary the number of people in the recordings (i.e. low versus high population density).

We hypothesize that this negative interaction effect occurred because reminders of the presence of others trigger a behavioural response suitable to a larger group of people sharing a resource pool. Humans are capable of responding to subtle cues in their environment that predict the likelihood that cooperation will be profitable (e.g. [16]). The people-chatting sound is known to invoke feelings of high population density [33], which typically indicates that a large group of people are sharing the same resources. It is known that increasing group size makes cooperation more difficult to achieve and a shared resource less likely to be sustained [26–28,57,58]. Therefore, if the motivation for cooperation is focused on a material payoff that is contingent on the resource being sustained (i.e. future stake), but larger group size reduces the likelihood that it will be sustained, then cues for larger group size (i.e. people-chatting sounds) decrease cooperation because cooperation is less likely to be profitable.

Our hypothesis was also supported by an analysis performed after the fact. If the negative interaction effect is due to the threat that larger groups pose to individual benefits when there is a future stake, then individualists should respond more strongly than prosocials. Although the results did not reach statistical significance, the effects were in the expected direction: the negative effect on cooperation of combining people-chatting sounds and future stakes was stronger for individualists than prosocials, and this was true even though having a future stake had a stronger positive effect for individualists than prosocials without people-chatting sounds (electronic supplementary material, figure S1).

The future-stakes scenario can be contrasted with the no-future-stakes scenario, where the only payoffs from being cooperative are non-material and, therefore, may be satisfied regardless of whether or not the pool is sustained. For example, if people-chatting sound also invokes reputational concerns, then provided that observers place higher importance on intentions than utilitarian efficacy [20,59],

cooperation will still provide a reputation payoff regardless of whether or not the pool was sustained. We hypothesize that this is why people-chatting sounds increased cooperation when there was no future stake but decreased it when there was a future stake. Introducing a future material stake crowded out [60] or shifted participants' goals away from non-material incentives, and thus cooperation declined in response to reminders of the presence of others because the presence of others made the material incentives less likely.

A theoretical model (electronic supplementary material, note C) supports the idea that, although future stakes can increase cooperation, the presence of others will have an opposing effect in an intergenerational context. In an evolutionary niche-construction model [8,9,61], if offspring are likely to use the same resource as their parents, then the direct cost of reducing resource extraction to an individual can be outweighed by future benefits to their descendants. Importantly, the only 'incentive' to behave sustainably in this model is a genetic future stake, and natural selection favours sustainable resource use via one's own descendants inheriting the resource (i.e. genealogical self-interest). Therefore, the model predicts that introducing a future stake increases cooperation. However, the model also makes a second prediction that is consistent with our results: as the number of individuals sharing the pool increases, cooperation declines.

Taken together, our findings lend support to the idea that environmental messages that invoke concern for 'our children' may backfire; however, more studies are needed to establish the result. Environmental messaging that appeals to self-interest, including one's genealogical self-interest, has been criticized on the grounds that the scope of ethical concern is inadequate to address large-scale environmental problems like climate change [7,15]. Our results also suggest that emphasizing an individual's future stake in a resource may backfire when the sustainability of that resource is dependent on the cooperation of a large number of people. However, a research gap remains when translating from an individual's material to genetic self-interest. While our treatment, with a 50% chance to inherit the pool in the next generation, is mathematically equivalent to the 50% genetic relatedness between a parent and offspring in theoretical models, it is unlikely that our experiment captured the true psychology of kinship. To better mirror the theoretical model, experiment work is needed using actual parent–child dyads. In addition, our game participants were all students in Singapore, and behaviour in experimental games is known to vary culturally and demographically [62]. Therefore, we recommend that future studies could investigate intergenerational games in different cultures and particularly with a wider range of age groups, who may respond differently to concerns for future generations.

## 4.3. The dynamic game also shows a negative interaction between future stakes and presence of others

Both of the key results from the one-shot game—the positive effect of a future stake on cooperation and the negative interaction between reminders of the presence of others and future stakes—were also observed in the dynamic game (figure 1). However, in the dynamic game, participants extracted more resource, which removed the effects in the later rounds and weakened the overall effects (electronic supplementary material, figure S2). Higher extraction in the dynamic game was consistent with our expectation that individuals would adjust their behaviour in response to observations of others (i.e. a spread of selfishness). Observing others over-exploit the resource reduces a cooperator's motivation to continue cooperating, particularly when the resource is likely to pass the over-exploitation threshold (approximately half of the groups passed the threshold at the third round).

## 5. Conclusion

Our results suggest that having a material stake in the future of a resource pool promotes intergenerational cooperation. However, the positive effects are reduced or even reversed when individuals are also reminded of the presence of others, which we interpret as a response to perceived increase in the number of others with access to the resource. Our result has implications for sustainability messaging that deserve further research. Many modern environmental crises are global in scale, involving billions of others. Sustainability is often framed in terms of protecting the environment for 'our children', which may be interpreted in terms of a material future stake (i.e. genealogical self-interest). If individuals feel the scale of cooperation required to solve the environmental problem is too large, then our results suggest that appealing to individuals' concern for their own children in this way may be counterproductive.

Ethics. The experiments were approved by the Institutional Review Board of National University of Singapore (IRB S-19-266).

Data accessibility. Experimental data are deposited in figshare at https://doi.org/10.6084/m9.figshare.12905753.

Authors' contributions. L.R.C., C.-c.C., T.P.L.N. and C.L.Y.T. designed experiments. N.P.K. derived theoretical models. C.-c.C., T.P.L.N. and C.L.Y.T. performed experiments. C.-c.C. analysed the data. C.-c.C., N.P.K. and L.R.C. wrote and all authors revised the manuscript.

Competing interests. The authors declare no competing interests.

Funding. We acknowledge research funds from the National Parks Board and the Ministry of National Development, Singapore.

Acknowledgements. We thank Ryan Chisholm for feedback on the manuscripts and Yuchen Zhang for experiment assistance.

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
