## [Peer Review File · Royal Society Open Science]

Review History

RSOS-210206.R0 (Original submission)

Review form: Reviewer 1

Is the manuscript scientifically sound in its present form?

Yes

Are the interpretations and conclusions justified by the results?

Yes

Is the language acceptable?

Yes

Do you have any ethical concerns with this paper?

No

Have you any concerns about statistical analyses in this paper?

No

Recommendation?

Accept with minor revision (please list in comments)

Comments to the Author(s)

In this paper, the authors assess the roles of having a stake in the future and reminders of the presence of others on intergenerational cooperation by combining experimental and theoretical approaches. First, participants were told that there was a 50% chance that they would inherit the pool in the next generation of the game. Then, a sound-treatment was used to explore the effect of the presence of others. The main message is brought across fully supported by the presented results. I have enjoyed reading this paper. However, it must be carefully revised before being published and the following points should be considered:

1. Regarding the participants in the experiment, the sample lacks universality because participants are all from NUS and most of them are Chinese ethnicity (87%). Moreover, participants are asked to provide demographic information, but they were not analysed detailedly in the sections of Results Discussion. (This part is just mentioned in Line 240 -243)
2. Obviously, the points in Figure 1 are very discrete, and the prediction lines have no real meaning.
- 3 The English style and grammar should be improved as much as possible.

Review form: Reviewer 2

Is the manuscript scientifically sound in its present form?

Yes

Are the interpretations and conclusions justified by the results?

Yes

Is the language acceptable?

Yes

Do you have any ethical concerns with this paper?

No

Have you any concerns about statistical analyses in this paper?

No

Recommendation?

Accept as is

Comments to the Author(s)

I have previously reviewed an earlier version of this manuscript at a different journal.

Since then the authors have done a major rewrite and did a great job addressing my concern. What used to be interpreted as relatedness is now (correctly, as I believe) interpreted as having a stake in the future. There is also additional analyses and the paper has generally been streamlined a bit more, with the speculations of what might be driving these effects now in the

discussion. I believe all of these changes have improved the manuscript and I thank the authors for taking these comments on board.

I have no other suggestions to make and look forward to seeing this paper published.

Review form: Reviewer 3

Is the manuscript scientifically sound in its present form?

Yes

Are the interpretations and conclusions justified by the results?

No

Is the language acceptable?

Yes

Do you have any ethical concerns with this paper?

No

Have you any concerns about statistical analyses in this paper?

No

Recommendation?

Major revision is needed (please make suggestions in comments)

Comments to the Author(s)

The current manuscript presents an interesting study that examines how exposure to simulated dense conditions can influence cooperation in public goods games. What is particularly interesting are the effects on “intergenerational resource” games. The study finds that even though people are more cooperative when told that they might “inherit” the outcomes of their decisions, this effect is attenuated or even reversed when social density is made salient. I think the work is interesting and presents meaningful boundary conditions for social cooperation. That said, there are several non-trivial issues with the study methods and interpretation of the findings here.

A major methodological issue is the use of nature sounds as a “control” sound condition. The authors’ interpretation of the current findings is that perceptions of larger group size, as induced by the people sounds manipulation, led to less cooperation. And this is because the crowd sound manipulation led to participants thinking that there are more people who will be sharing the pool, which is a less sustainable situation. If so, an appropriate control would actually have been a sound manipulation with FEWER voices, or just one that would seem less crowded in general. The nature sound control has no people in it at all. Coupled with the fact that there is existing work finding that exposure to nature stimuli may increase prosocial behavior and reduce materialism (see references below), this makes interpretation of the current findings less clear.

Piff, P. K., Dietze, P., Feinberg, M., Stancato, D. M., & Keltner, D. (2015). Awe, the small self, and prosocial behavior. *Journal of personality and social psychology*, 108(6), 883.

Joye, Y., Bolderdijk, J. W., Köster, M. A., & Piff, P. K. (2020). A diminishment of desire: Exposure to nature relative to urban environments dampens materialism. *Urban Forestry & Urban Greening*, 54, 126783.

It would be ideal to repeat the study with a people sounds manipulation that really indicates a smaller group size. If not, there might also be other ways to help clarify the interpretation. For instance, if the people sound effects are indeed due to perceptions of larger groups, which pose a threat to individual benefits, might we expect specifically people with “individualist” social value orientations to react to the people sound manipulation? In other words, there might be an interaction between individual differences such as SVO and the experimental manipulations. In general, I think more needs to be done before I’m convinced of the current interpretation of the crowd sound effects.

In relation to this, the authors also note the unexpected finding of people being exposed to the crowd sound becoming more cooperative, when there was no future stake. Past work using the sound manipulation, which the authors cite (reference 33), provides a relevant explanation for this. That work found that exposure to the crowd sound led individuals to become more future oriented and to delay gratification on a financial decision. It would be important to discuss the current findings in relation to prior work, given the conceptual connections.

A second concern I had was the nature of the intergenerational common-pool resource game used here. The manuscript frequently frames this in terms of passing on the resource pool to one’s offspring (e.g., 50% chance means 50% genes as shared with one’s offspring). Given the manipulation instructions presented in the supplementary notes, this seems a stretch. If anything, the future stakes manipulation seems to be simply a manipulation that makes individuals accountable for their OWN decisions (they will inherit the consequences of their own resource use). A “true” intergenerational game would have brought in actual parent-child dyads and have the parent pass their resource pool on to their children when it’s their child’s turn to play the game. The current design just doesn’t seem to be able to speak to people’s psychology of kinship, and the authors need to avoid that overinterpretation.

Decision letter (RSOS-210206.R0)

Dear Ms Chang

The Editors assigned to your paper RSOS-210206 "Having a stake in the future and perceived population density influence intergenerational cooperation" have now received comments from reviewers and would like you to revise the paper in accordance with the reviewer comments and any comments from the Editors. Please note this decision does not guarantee eventual acceptance.

We invite you to respond to the comments supplied below and revise your manuscript. Below the referees’ and Editors’ comments (where applicable) we provide additional requirements. Final acceptance of your manuscript is dependent on these requirements being met. We provide guidance below to help you prepare your revision.

Please submit your revised manuscript and required files (see below) no later than 21 days from today's (ie 31-Mar-2021) date. Note: the ScholarOne system will 'lock' if submission of the revision is attempted 21 or more days after the deadline. If you do not think you will be able to meet this deadline please contact the editorial office immediately.

on behalf of Professor Matjaz Perc (Associate Editor) and Pete Smith (Subject Editor)
openscience@royalsociety.org

Reviewer comments to Author:

Reviewer: 1

Comments to the Author(s)

In this paper, the authors assess the roles of having a stake in the future and reminders of the presence of others on intergenerational cooperation by combining experimental and theoretical approaches. First, participants were told that there was a 50% chance that they would inherit the pool in the next generation of the game. Then, a sound-treatment was used to explore the effect of the presence of others. The main message is brought across fully supported by the presented results. I have enjoyed reading this paper. However, it must be carefully revised before being published and the following points should be considered:

1. Regarding the participants in the experiment, the sample lacks universality because participants are all from NUS and most of them are Chinese ethnicity (87%). Moreover, participants are asked to provide demographic information, but they were not analysed detailly in the sections of Results Discussion. (This part is just mentioned in Line 240 -243)
2. Obviously, the points in Figure 1 are very discrete, and the prediction lines have no real meaning.
- 3 The English style and grammar should be improved as much as possible.

Reviewer: 2

Comments to the Author(s)

I have previously reviewed an earlier version of this manuscript at a different journal.

Since then the authors have done a major rewrite and did a great job addressing my concern. What used to be interpreted as relatedness is now (correctly, as I believe) interpreted as having a

stake in the future. There is also additional analyses and the paper has generally been streamlined a bit more, with the speculations of what might be driving these effects now in the discussion. I believe all of these changes have improved the manuscript and I thank the authors for taking these comments on board.

I have no other suggestions to make and look forward to seeing this paper published.

Reviewer: 3

Comments to the Author(s)

The current manuscript presents an interesting study that examines how exposure to simulated dense conditions can influence cooperation in public goods games. What is particularly interesting are the effects on “intergenerational resource” games. The study finds that even though people are more cooperative when told that they might “inherit” the outcomes of their decisions, this effect is attenuated or even reversed when social density is made salient. I think the work is interesting and presents meaningful boundary conditions for social cooperation. That said, there are several non-trivial issues with the study methods and interpretation of the findings here.

A major methodological issue is the use of nature sounds as a “control” sound condition. The authors’ interpretation of the current findings is that perceptions of larger group size, as induced by the people sounds manipulation, led to less cooperation. And this is because the crowd sound manipulation led to participants thinking that there are more people who will be sharing the pool, which is a less sustainable situation. If so, an appropriate control would actually have been a sound manipulation with FEWER voices, or just one that would seem less crowded in general. The nature sound control has no people in it at all. Coupled with the fact that there is existing work finding that exposure to nature stimuli may increase prosocial behavior and reduce materialism (see references below), this makes interpretation of the current findings less clear.

Piff, P. K., Dietze, P., Feinberg, M., Stancato, D. M., & Keltner, D. (2015). Awe, the small self, and prosocial behavior. *Journal of personality and social psychology*, 108(6), 883.

Joye, Y., Bolderdijk, J. W., Köster, M. A., & Piff, P. K. (2020). A diminishment of desire: Exposure to nature relative to urban environments dampens materialism. *Urban Forestry & Urban Greening*, 54, 126783.

It would be ideal to repeat the study with a people sounds manipulation that really indicates a smaller group size. If not, there might also be other ways to help clarify the interpretation. For instance, if the people sound effects are indeed due to perceptions of larger groups, which pose a threat to individual benefits, might we expect specifically people with “individualist” social value orientations to react to the people sound manipulation? In other words, there might be an interaction between individual differences such as SVO and the experimental manipulations. In general, I think more needs to be done before I’m convinced of the current interpretation of the crowd sound effects.

In relation to this, the authors also note the unexpected finding of people being exposed to the crowd sound becoming more cooperative, when there was no future stake. Past work using the sound manipulation, which the authors cite (reference 33), provides a relevant explanation for this. That work found that exposure to the crowd sound led individuals to become more future oriented and to delay gratification on a financial decision. It would be important to discuss the current findings in relation to prior work, given the conceptual connections.

A second concern I had was the nature of the intergenerational common-pool resource game used here. The manuscript frequently frames this in terms of passing on the resource pool to one's offspring (e.g., 50% chance means 50% genes as shared with one's offspring). Given the manipulation instructions presented in the supplementary notes, this seems a stretch. If anything, the future stakes manipulation seems to be simply a manipulation that makes individuals accountable for their OWN decisions (they will inherit the consequences of their own resource use). A "true" intergenerational game would have brought in actual parent-child dyads and have the parent pass their resource pool on to their children when it's their child's turn to play the game. The current design just doesn't seem to be able to speak to people's psychology of kinship, and the authors need to avoid that overinterpretation.

===PREPARING YOUR MANUSCRIPT===

===PREPARING YOUR REVISION IN SCHOLARONE===

Author's Response to Decision Letter for (RSOS-210206.R0)

See Appendix A.

RSOS-210206.R1 (Revision)

Review form: Reviewer 1

Is the manuscript scientifically sound in its present form?

Yes

Are the interpretations and conclusions justified by the results?

Yes

Is the language acceptable?

Yes

Do you have any ethical concerns with this paper?

No

Have you any concerns about statistical analyses in this paper?

No

Recommendation?

Accept as is

Comments to the Author(s)

The authors have addressed the limitations pointed out by my last review report, and they have also answered all the questions in detail.

I have no hesitation to recommend the article to be published in the current version.

Review form: Reviewer 3

Is the manuscript scientifically sound in its present form?

Yes

Are the interpretations and conclusions justified by the results?

Yes

Is the language acceptable?

Yes

Do you have any ethical concerns with this paper?

No

Have you any concerns about statistical analyses in this paper?

No

Recommendation?

Accept as is

Comments to the Author(s)

The authors have done an excellent job in their revision. They have responded thoughtfully to all the issues I raised in my review. I am satisfied by all the responses, and am persuaded by the authors' defense of their methods and interpretation. Well done! I look forward to seeing the work published.

Decision letter (RSOS-210206.R1)

Dear Ms Chang,

It is a pleasure to accept your manuscript entitled "Having a stake in the future and perceived population density influence intergenerational cooperation" in its current form for publication in Royal Society Open Science. The comments of the reviewer(s) who reviewed your manuscript are included at the foot of this letter.

on behalf of Pete Smith (Subject Editor)
openscience@royalsociety.org

Reviewer comments to Author:

Reviewer: 1

Comments to the Author(s)

The authors have addressed the limitations pointed out by my last review report, and they have also answered all the questions in detail.

I have no hesitation to recommend the article to be published in the current version.

Reviewer: 3

Comments to the Author(s)

The authors have done an excellent job in their revision. They have responded thoughtfully to all the issues I raised in my review. I am satisfied by all the responses, and am persuaded by the authors' defense of their methods and interpretation. Well done! I look forward to seeing the work published.

Appendix A

Dear Professor Parkhouse,

Please find herewith a revised version of our manuscript entitled "Having a stake in the future and perceived population density influence intergenerational cooperation". We thank the reviewers for their comments. They were very insightful and allowed us to improve the article. We have addressed the two limitations pointed out by the reviewers, and we have also reorganized the discussion section of the manuscript to make the argument clearer.

We hope you agree with us that the article is now acceptable for publication.

Best Regards,

Chia-chen Chang

Reviewer: 1

Comment 1: In this paper, the authors assess the roles of having a stake in the future and reminders of the presence of others on intergenerational cooperation by combining experimental and theoretical approaches. First, participants were told that there was a 50% chance that they would inherit the pool in the next generation of the game. Then, a sound-treatment was used to explore the effect of the presence of others. The main message is brought across fully supported by the presented results. I have enjoyed reading this paper. However, it must be carefully revised before being published and the following points should be considered:

Response 1: Thank you for the feedback. We have revised the manuscript following the comments below.

Comment 2: Regarding the participants in the experiment, the sample lacks universality because participants are all from NUS and most of them are Chinese ethnicity (87%). Moreover, participants are asked to provide demographic information, but they were not analysed detailly in the sections of Results Discussion. (This part is just mentioned in Line 240 -243)

Response 2: Thank you for pointing this out. It is true that our participants were mostly Chinese ethnicity. This is because Chinese are the dominant ethnicity in Singapore (about 76% of Singapore citizens). We have also analyzed whether ethnicity is associated with their resource extraction behaviour, as well as other demographic factors. The results are now mentioned in Line 243 – 247.

Action in manuscript 2:

Result lines 237-242:

“For participant’s characteristics, groups having more prosocial members extracted fewer points, and prosocial individuals also extracted fewer points (Figure 2). Nature-related ~~and female~~ individuals extracted fewer points, and groups with more STEM-major members extracted fewer points (Figure 2). With regard to demographic factors, we found female individuals and groups with more female members extracted fewer points (Figure 2). Groups with members from higher income family extracted slightly more points (Figure 2). We did not detect a significant association between ethnicity and extraction behaviour, and we also did not detect a significant association between participants having taken environmental modules and their extraction behaviour (Figure 2).”

As to the broader point that our sample does not necessarily generalise to other countries and demographic groups, we agree. We are hopeful that, with our result in hand, future researchers will be able to obtain the funding required to do broader scope studies.

Discussion lines 360-365:

“In addition, our game participants were all Singaporean students, and behaviour in experimental games is known to vary culturally and demographically [62]. Therefore, we recommend that future studies could investigate intergenerational games in different cultures and particularly with a wider range of age groups, who may respond differently to concerns for future generations.”

Comment 3: Obviously, the points in Figure 1 are very discrete, and the prediction lines have no real meaning.

Response 3: Thank you for the comment. The lines do have meaning; they indicate the effect from the best fitting model while controlling for other variables in the model. The discrete points in the figure show the variability among participants. We find that the lines from the best model are important because they help visualize the effects of interests while controlling for the other variables. To make this clearer, we have now edited the figure legend.

Action in manuscript 3:

Additional information in the figure legend:

“Figure 1. Effect of future-stake treatment (without a future stake vs having a future stake) and the sound treatment (silence, nature, and people-chatting sounds) on the group extraction in one-shot game (a) and 5-rounds dynamic game (b). The plots include prediction lines of the best model while controlling for other variables, and discrete points indicate high variability among groups. The visualization was done using visreg package in R”

Comment 4: The English style and grammar should be improved as much as possible.

Response 4: Thank you for the suggestion. We have now further improved the English writing.

Reviewer: 2

Comment 1: I have previously reviewed an earlier version of this manuscript at a different

journal.

Since then the authors have done a major rewrite and did a great job addressing my concern. What used to be interpreted as relatedness is now (correctly, as I believe) interpreted as having a stake in the future. There is also additional analyses and the paper has generally been streamlined a bit more, with the speculations of what might be driving these effects now in the discussion. I believe all of these changes have improved the manuscript and I thank the authors for taking these comments on board.

I have no other suggestions to make and look forward to seeing this paper published.

Response 1: Thank you for the positive feedback and for the useful comments in the previous review that helped us improve the article.

Reviewer: 3

Comment 1: The current manuscript presents an interesting study that examines how exposure to simulated dense conditions can influence cooperation in public goods games. What is particularly interesting are the effects on “intergenerational resource” games. The study finds that even though people are more cooperative when told that they might “inherit” the outcomes of their decisions, this effect is attenuated or even reversed when social density is made salient. I think the work is interesting and presents meaningful boundary conditions for social cooperation. That said, there are several non-trivial issues with the study methods and interpretation of the findings here.

Response 1: Thank you for the encouraging and constructive feedback. We have now revised the manuscript following the comments below.

Comment 2: A major methodological issue is the use of nature sounds as a “control” sound condition. The authors’ interpretation of the current findings is that perceptions of larger group size, as induced by the people sounds manipulation, led to less cooperation. And this is because the crowd sound manipulation led to participants thinking that there are more people who will be sharing the pool, which is a less sustainable situation. If so, an appropriate control would actually have been a sound manipulation with FEWER voices, or just one that would seem less crowded in general. The nature sound control has no people in it at all. Coupled with the fact that there is existing work finding that exposure to nature stimuli may increase prosocial behavior and reduce materialism (see references below), this makes interpretation of the current findings less clear.

Piff, P. K., Dietze, P., Feinberg, M., Stancato, D. M., & Keltner, D. (2015). Awe, the small self, and prosocial behavior. *Journal of personality and social psychology*, 108(6), 883.

Joye, Y., Bolderdijk, J. W., Köster, M. A., & Piff, P. K. (2020). A diminishment of desire: Exposure to nature relative to urban environments dampens materialism. *Urban Forestry & Urban Greening*, 54, 126783.

Response 2:

Although nature sounds was not an ideal control, we disagree that nature sounds was not a useful control, and the fact that nature sounds alone also marginally increased cooperation when there is no future stake makes interpreting our findings much easier.

Both people-chatting sounds and nature sounds increased cooperation when there is no future stake, but it was only people-chatting sounds that undermined the positive effect of having future stakes, leading to decrease in cooperation -- and that is the key result of the paper. By comparing with nature sounds, we showed that the interaction does not occur because of sound in general, nor does it occur with another sound that also increased cooperation otherwise. Instead, it occurs specifically with people-chatting sounds, which previous work has shown to induce feelings of crowdedness, and which theory predicts will reduce cooperation in the situation we observed, i.e., crowdedness reduces cooperation when the payoff depends on material future stakes.

Nevertheless, we agree that future work using a sound manipulation that varies the number of people voices is an excellent suggestion, and we have added it to the text where we make other suggestions for future work (Lines 307-309). We have shown that crowdedness reduces cooperation when there are future stakes, future workers may wish to clarify the exact relationship between crowd size and the effect. We also thank the Reviewer for the suggested references. We have added both of their papers alongside the paper we had already cited [47, 48].

Action in manuscript 2:

Discussion Lines 307-309: “Future studies interested in quantifying the effects of crowdedness on the interaction effect could use soundtracks that vary the number of people in the recordings (i.e., low versus high population density).”

Comment 3: It would be ideal to repeat the study with a people sounds manipulation that really indicates a smaller group size. If not, there might also be other ways to help clarify the interpretation. For instance, if the people sound effects are indeed due to perceptions of larger groups, which pose a threat to individual benefits, might we expect specifically people with “individualist” social value orientations to react to the people sound manipulation? In other words, there might be an interaction between individual differences such as SVO and the experimental manipulations. In general, I think more needs to be done before I’m convinced of the current interpretation of the crowd sound effects.

Response 3: We are very grateful to Reviewer 3 for suggesting this analysis, which has strengthened our argument. Although our result did not reach statistical significance (p-value cut-off 0.05), the effects were in the direction predicted by Reviewer 3 to be consistent with our argument. We now have added these in the discussion (lines 321-327). Specifically, we found that having a future stake had a stronger impact on individualistic than prosocial participants,

and the negative interaction effect of people-chatting sounds and future stakes was also stronger for individualistic than prosocial participants.

Action in manuscript 3:

Discussion Lines 321-327: “Our hypothesis was also supported by analysis performed after the fact. If the negative interaction effect is due to the threat that larger groups pose to individual benefits when there is a future stake, then individualists should respond more strongly than prosocials. Although the results did not reach statistical significance, the effects were in the expected direction: the negative effect on cooperation of combining people-chatting and future stakes was stronger for individualists than prosocials, and this was true even though having a future stake had a stronger positive effect for individualists than prosocials without people-chatting sounds (Figure S1).”

Figure from S1:

Comment 4: In relation to this, the authors also note the unexpected finding of people being exposed to the crowd sound becoming more cooperative, when there was no future stake.

Response 4: Reviewer 3 has drawn our attention to the fact that "unexpected result" was a poor/distracting choice of phrasing to use. First, we must correct the reviewer: the unexpected result was not the finding that people exposed to the people-chatting sound became more cooperative when there was no future stake. Indeed, we expected that result based on the reputation effect (introduction lines 45-50). The unexpected result was the finding that there was no significant difference between people-chatting sounds and nature sounds. However, as Reviewer 3 points out above, and as we discuss later in this paragraph (Lines 274-288), that lack

of difference was not entirely unexpected because nature stimuli is also known to increase prosocial behaviour. To avoid misunderstanding, we have removed that phrase and rewritten the paragraph.

Action in manuscript 4:
Discussion lines 267-288:

“When there is no future stake, reminders of the presence of others could increase cooperation

When participants had ~~there was no~~ future stake in the future of the pool, ~~we observed an increase in cooperation when participants were exposed to~~ people-chatting sounds compared to silence. This positive effect was expected because people-chatting sounds invoke the presence of others [41], which may provide a subtle cue to trigger the reputation effect [16], similar to the watching eye effect [20]. Alternatively, people-chatting sounds have also been shown to increase future orientation of individuals [33], which would also reduce the resource extraction.

~~;~~ however, ~~w~~We found no significant difference between people-chatting sounds and nature sounds, likely because nature sounds also had a marginally positive effect on cooperation compared to silence, ~~and nature sounds had a marginally positive effect on cooperation compared to silence. This was contrary to our expectation that the people-chatting sounds would increase cooperation compared to nature sounds. We expected people-chatting sounds to invoke the presence of others [52], which would provide a subtle cue to trigger the reputation effect [16], similar to the watching eye effect [20]. One possible explanation for our unexpected result is that sound itself increased cooperation. Previous work has found that increasing the cognitive load on participants, e.g., with distracting sounds [4253], causes them to rely on an intuitive cooperative behaviour [4354] and promotes future-oriented and other-oriented behaviour [4455, 4556]. An alternative explanation is that nature sounds specifically increase cooperation through a different mechanism. Combining nature sounds with a game scenario that has an obvious sustainability analogy may have activated pro-environmental values in some participants. For example, it has been found that people behave more sustainably and cooperatively after experiencing nature [4657], e.g., due to reduction of materialism and increase in generosity through the experience of awe from nature [47, 48], and people with higher nature relatedness also have stronger pro-environmental attitudes in general [37, 4958]. In support of this, we also found that more nature-related participants extracted less resources. However, we did not find a group level correlation between nature relatedness and cooperation, which is likely due to low variation across groups (group scores ranging from 2.96 to 3.98 and individual scores ranging from 1.76 to 4.62). Future studies could use music or white noise to isolate the effects specific to people or nature sounds from sound in general~~ further test the effect of sounds.”

Comment 5: Past work using the sound manipulation, which the authors cite (reference 33), provides a relevant explanation for this. That work found that exposure to the crowd sound led individuals to become more future oriented and to delay gratification on a financial decision. It would be important to discuss the current findings in relation to prior work, given the conceptual connections.

Response 5:

Unfortunately, Ref. 33 (Sng et al.) does not provide a suitable explanation for that result because that would require applying Sng et al.'s concept of 'future orientation' beyond the scope of the original theory, and doing so causes Sng et al.'s theory to predict the opposite of what we observed across future-stakes treatments.

Sng et al. hypothesised that crowdedness induced future orientation because crowdedness is a cue for competition, and when competition between humans is more intense, future orientation increases evolutionary fitness. As Reviewer 3 mentioned, Sng et al. observed that individuals exposed to crowd sounds had a greater preference for higher delayed rewards. That result was one of a series showing that crowdedness was associated with greater future orientation in general, including: later marriage, fewer children, and higher investment in education for one's self and one's children. They explained these observations in terms of life-history theory, which is about how the timing of activities and resources allocation can be plastic to maximises fitness given a particular environmental context. For example, if the environmental context presents a high (low) mortality risk, then natural selection favours a fast (slow) life-history strategy with earlier (later) reproduction and a larger (smaller) number of lower-quality (high-quality) offspring. In the context of humans, the competitive environment is most often hierarchical: success is determined by the amount of resources and embodied capital that an individual can accumulate and invest in their children. Therefore, their observations are consistent with humans having a slow life-history strategy; humans respond to cues of increased competition by foregoing immediate rewards to invest in a later higher reproductive payoff.

It is important to note that, in contrast to our experiments, all of the future oriented behaviours observed by Sng et al. are "selfish"; they affect one's future self and/or genetic offspring. The theory does not apply to more generalised future concerns such the impact on the natural resources or on future unrelated generations. Likewise, the theoretical rationale for the shift in time perspective is grounded in evolutionary theory, which is directed at maximising the individual's inclusive fitness, not the fitness of others. Therefore, if we are to apply Sng's theory to our result as Reviewer 3 suggests, then we must broaden the scope of 'future orientation' to include concern for future unrelated others. This is necessary because the result that Reviewer 3 mentioned (crowd sounds reduced resource extraction) occurred when participants had no material stake in the future of the pool.

Further, if we broaden the scope of Sng's 'future orientation' to apply it to our results across future-stakes treatments, then it predicts the opposite of what we observed. We would expect crowdedness to improve cooperation especially when there is a future stake. Instead, our results show that the positive effect of crowd sounds on cooperation was more obvious when participants had no future stake in the pool; and when there was a future stake, crowd sounds had a negative effect on cooperation (Fig. 1).

In contrast to future-orientation explanation above, reputation explains why crowd sounds increased cooperation when there was no future stake without going beyond the original scope of the theory (discussion lines 268-273 and lines 328-337). That is not to say that we rule out the possibility that crowd sounds induced a generalised future orientation that included deeper

consideration for the future of the pool. Rather, there is nothing in this possibility that predicts or explains why crowd sounds increased resource extraction when there is a future stake, which is the treatment where the Sng's theory should be the most applicable. A more likely explanation is that introducing future stakes focused participants' minds on the future material benefits of the pool, which are less likely to be preserved when many people are sharing the same resources (discussion lines 310-320).

Nevertheless, we agree that Sng et al.'s work has important conceptual connections to our own, and we have added some text highlighting an additional connection with that work in the discussion.

Action in manuscript 5:

Discussion lines 269-273: “This positive effect was expected because people-chatting sounds invoke the presence of others [41], which may provide a subtle cue to trigger the reputation effect [16], similar to the watching eye effect [20]. Alternatively, people-chatting sounds have also been shown to increase future orientation of individuals [33], which would also reduce the resource extraction.”

Comment 6: A second concern I had was the nature of the intergenerational common-pool resource game used here. The manuscript frequently frames this in terms of passing on the resource pool to one's offspring (e.g., 50% chance means 50% genes as shared with one's offspring). Given the manipulation instructions presented in the supplementary notes, this seems a stretch. If anything, the future stakes manipulation seems to be simply a manipulation that makes individuals accountable for their OWN decisions (they will inherit the consequences of their own resource use). A “true” intergenerational game would have brought in actual parent-child dyads and have the parent pass their resource pool on to their children when it's their child's turn to play the game. The current design just doesn't seem to be able to speak to people's psychology of kinship, and the authors need to avoid that overinterpretation.

Response 6:

We agree that we must avoid over-interpreting our result in terms of kinship. Our theoretical model and results combined strongly hint that kinship is a worthwhile avenue for future work. But to make clear that it is a suggestion rather than a direct conclusion from our work, we have rewritten these perspectives in the Discussion (see also Reviewer 2's positive comments about us using the Discussion in this way).

Action in manuscript 6:

Discussion lines 349-365: “Taken together, our findings lend support to the idea that environmental messages that invoke concern for “our children” may backfire; however, more studies are needed to establish the result. Environmental messaging that appeals to self-interest, including one's genealogical self-interest, has been criticised on the grounds that the scope of ethical concern is inadequate to address large-scale environmental problems like climate change [7,15]. Our results also suggest that emphasising an individual's future stake in a resource may

backfire when the sustainability of that resource is dependent on the cooperation of a large number of people. However, a research gap remains when translating from an individual's material to genetic self-interest. While our treatment, with a 50% chance to inherit the pool in the next generation, is mathematically equivalent to the 50% relatedness between parent and offspring in theoretical models, it is unlikely that our experiment captured the true psychology of kinship. To better mirror the theoretical model, experiment work is needed using actual parent-child dyads. In addition, our game participants were all Singaporean students, and behaviour in experimental games is known to vary culturally and demographically [62]. Therefore, we recommend that future studies could investigate intergenerational games in different cultures and particularly with a wider range of age groups, who may respond differently to concerns for future generations.”